# Kinetics of the xanthophyll cycle and its role in photoprotective memory and response

Audrey Short[1,2,3,9], Thomas P. Fay[4,9], Thien Crisanto[2,5,6], Ratul Mangal [4], Krishna K. Niyogi [2,5,6], David T. Limmer [3,4,7,8] & Graham R. Fleming [1,2,3,4] ✉

Efficiently balancing photochemistry and photoprotection is crucial for survival and productivity of photosynthetic organisms in the rapidly fluctuating light levels found in natural environments. The ability to respond quickly to sudden changes in light level is clearly advantageous. In the alga *Nannochloropsis oceanica* we observed an ability to respond rapidly to sudden increases in light level which occur soon after a previous high-light exposure. This ability implies a kind of memory. In this work, we explore the xanthophyll cycle in *N. oceanica* as a short-term photoprotective memory system. By combining snapshot fluorescence lifetime measurements with a biochemistry-based quantitative model, we show that short-term memory arises from the xanthophyll cycle. In addition, the model enables us to characterize the relative quenching abilities of the three xanthophyll cycle components. Given the ubiquity of the xanthophyll cycle in photosynthetic organisms the model described here will be of utility in improving our understanding of vascular plant and algal photoprotection with important implications for crop productivity.

In high-intensity light, photosynthetic organisms are unable to utilize all available energy for photochemistry. In order to minimize the formation of damaging reactive oxygen species, the excess energy is dissipated as heat through non-photochemical quenching (NPQ) pathways[1,2]. The eustigmatophyte alga *Nannochloropsis oceanica* has a relatively simple NPQ system[3,4] in comparison to vascular land plants. It consists of two main components: a pH-sensing protein, potentially LHCX1, and the xanthophyll cycle. The xanthophyll cycle in *N. oceanica* is a shared feature with higher plants, but this alga lacks additional features like state transitions or pigments like lutein and chlorophyll-*b*[5–7]. This simplistic nature makes *N. oceanica* an ideal model organism for studying the essential components of NPQ.

The xanthophyll cycle in *N. oceanica* consists of the same de-epoxidation steps, from violaxanthin (V) to antheraxanthin (A) to zeaxanthin (Z), and reverse epoxidation steps, as seen in green algae and plants[6,8]. The enzyme violaxanthin de-epoxidase (VDE), located in the thylakoid lumen, converts V to A to Z upon protonation under high-light (HL) stress. Simultaneously, zeaxanthin epoxidase (ZEP), which is found in the stroma and thought to be constitutively active, reverses the VAZ cycle by epoxidizing Z to A to V[9–11] (Fig. 1). It is now well-established that the VAZ cycle correlates with activation of energy-dependent quenching, qE, in both *N. oceanica*[3,4] and more complex organisms[12–14]. The fast activating, pH-dependent quenching, qE, in *N. oceanica* also depends on the protein LHCX1[4]. The mechanism of sensing changes in the thylakoid membrane pH-gradient and whether or not LHCX1 can bind pigments is still under investigation[4,15–19], however, the vital role of Z together with a pH-sensing protein in qE is widely achknowledged[8,14]. The accumulation of A and Z has been

[1]Graduate Group in Biophysics, University of California, Berkeley, CA 94720, USA. [2]Molecular Biophysics and Integrated Bioimaging Division Lawrence Berkeley National Laboratory, Berkeley, CA 94720, USA. [3]Kavli Energy Nanoscience Institute, Berkeley, CA 94720, USA. [4]Department of Chemistry, University of California Berkeley, Berkeley, CA 94720, USA. [5]Department of Plant and Microbial Biology, University of California, Berkeley, CA 94720, USA. [6]Howard Hughes Medical Institute, University of California, Berkeley, CA 94720, USA. [7]Chemical Science Division Lawrence Berkeley National Laboratory, Berkeley, CA 94720, USA. [8]Material Science Division Lawrence Berkeley National Laboratory, Berkeley, CA 94720, USA. [9]These authors contributed equally: Audrey Short, Thomas P. Fay. ✉e-mail: grfleming@lbl.gov

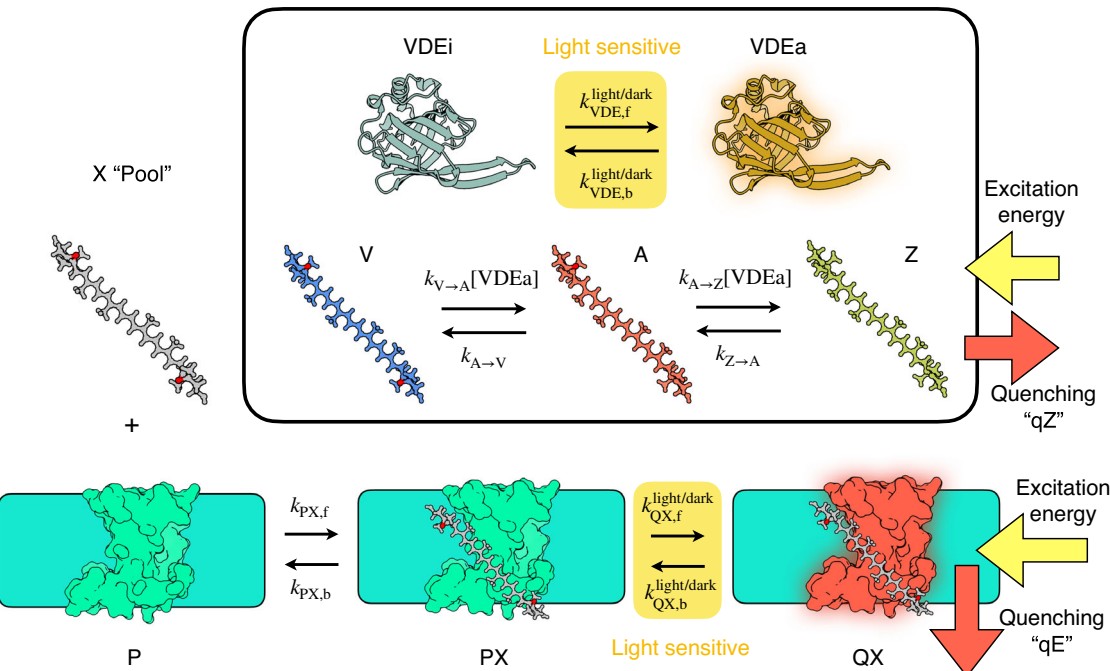

**Fig. 1 | Illustration showing the processes included in the xanthophyll cycle-based model.** The xanthophyll (X) binds to the protein (P) reversibly to form a protein-xanthophyll complex (PX). In response to light this can convert into an active quencher form (QX). When not bound to the protein, the xanthophylls interconvert between violaxanthin (V), antheraxanthin (A) and zeaxanthin (Z). The activation of the VDE enzyme, which controls the V → A → Z processes, is dependent on light conditions, which alter the ratio of the active VDE enzyme (VDEa) and its inactive from (VDEi). The light-sensitive steps in the model are highlighted in yellow. The species responsible for quenching, the QX complexes in qE and pool Z in qZ, are also indicated by red arrows.

observed to correlate with an increase in NPQ throughout a diurnal cycle in plants[10,11], and it has been proposed that an additional, slower activating and slow deactivating Z-dependent quenching process, also operates in the absence of a pH-gradient sensing protein[12,13]. However, the precise roles of the three xanthophylls and the kinetics of their interconversion in NPQ are not well understood, which is surprising given the prevalence of this widespread three-state photoprotective system in photosynthetic organisms.

In previous work[20], we utilized a simplified kinetic model of the VAZ cycle that did not include the intermediate A to understand NPQ in *N. oceanica*. Despite this simplification, the model gave useful insights into the time scales of processes involved in NPQ activation, and it could quantitatively predict the quenching response, as well as qualitatively predict changes in V and Z concentrations, in response to a variety of regular and irregular light/dark illumination sequences. However, when exploring how the response changed when the dark period was progressively lengthened, it became clear that *N. oceanica* has short-term "memory" of previous HL exposure which could not be captured by the simplified two-xanthophyll model. This type of memory of previous exposure to stressor events, wherein some organisms remain primed for an extended period to quickly respond to further stress, has been observed for other stressors such as in drought conditions[21]. Various plant species, including *Smilax australis*, *Monstera deliciosa*, *Vinca minor,* and *Vinca major*, have been shown to possess a long-term memory of growth light conditions, which is strongly species-dependent. This long-term memory manifests in xanthophyll pool size and composition as well as maximum NPQ levels[22,23], an effect we also found evidence for previously in *N. oceanica*[20]. It has also been shown that in phytoplankton and algae possessing a simpler two-state xanthophyll cycle, the xanthophylls can act as a long-term memory of growth light conditons[24–26]. In this work we aim to explore the details of short-term photoprotective memory

(operating on time scales ≲ 1 hour), complementing existing studies on connections between longer-term light exposure memory and the xanthophyll cycle.

We hypothesize that in response to light stress, the VAZ cycle, and the kinetics of the different de-epoxidation/epoxidation steps, may act as a memory of previous HL exposure[27]. Specifically, we propose that the presence of A in a system could keep plants and algae primed to respond to further HL stress, due to the slow rates of transforming A back to V. The role of the partially de-epoxidised xanthophyll A in photoprotection has been difficult to investigate directly, however, work on plants has suggested that both A and Z correlate with NPQ in plants[22,28], but in this work, we also aim to further elucidate its role in photoprotection. Previous work has shown the ratio of the rates from A → Z to V → A ranges from 4.5–6.3 times faster in various plant species,[29–31] and the rate of epoxidation has been measured to be 1.4 times faster for Z than A[11]. However precise measurements of these rates in *N. oceanica* and their functional significance in NPQ and short-term memory of light stress have not been fully explored.

In this work, we aim to fully understand the role of xanthophyll cycle kinetics in photoprotective memory by considering the full VAZ cycle in modeling NPQ, and we show that differential rates of interconversion between the three xanthophylls are responsible for the multiple time-scales of photoprotective memory. In a further step towards a comprehensive understanding of NPQ in *N. oceanica*, the full VAZ model allows us to assess the relative quenching abilities of the three xanthophylls in the qE process, estimate the relative abundance of quenchers in the thylakoid membrane, and also quantify the relative contributions of LHCX1-dependent qE quenching and zeaxanthin-dependent qZ quenching in NPQ. In what follows, we start by briefly presenting our expanded model, then show how it accurately describes the HL stress responses of *N. oceanica*, and how it encodes the functional role of the VAZ cycle in photoprotection.

## Results

### Kinetic model of xanthophyll-mediated photoprotection

Motivated by measurements of xanthophyll concentrations and NPQ in response to light exposure (as presented in the next section), we have developed a new model for the coupled LHCX1-xanthophyll cycle photoprotection system in *N. oceanica*, as is summarized schematically in Fig. 1. Before presenting any results, we briefly summarize the features of the model (details of the kinetic equations are given in the SI). In the predecessor to this model[20], we neglected several important features that are included in the new model presented here, such as the intermediate A, which we will show plays an essential role in photoprotective memory, and the capability of each xanthophyll to act as a quencher, facilitated by LHCX1, which will be important for understanding the immediate response of *N. oceanica* to light stress. Furthermore, we will show that the new model can quantitatively describe xanthophyll concentrations in cells, enabling us to estimate the absolute abundance of quenching sites in *N. oceanica* and estimate its absolute quenching rate.

Overall the model involves 12 chemical species: the protein P, the three "pool" xanthophylls X = V, A, and Z, three xanthophyll-bound complexes PX in the non-quenching state and three in the quenching state QX, and the active (protonated) VDEa and inactive (unprotonated) VDEi forms of the VDE enzyme. Within the model, the protein P, binds the xanthophylls, X = V, A, Z, reversibly to form a complex PX. For simplicity, we assume a single labile xanthophyll binding site per P, which we have found is sufficient to interpret the available experimental data. This PX complex is activated under HL conditions to reversibly form an active quencher, establishing the PX⇌QX equilibrium, which we assume arises due to protonation and conformational changes. Previous work has identified LHCX1 as an essential component in activating the protein P, in the "qE" quenching mechanism[4,11,20], although the actual active quencher PX/QX could involve other proteins, especially since it is not known if LHCX1 binds pigments, and alternatively, LHCX1 may just induce the conformational changes in P to activate quenching. Thus the precise identity of PX/QX is open to interpretation. The total fluorescence decay rate $\tau_F(t)^{-1}$ of chlorophylls in the membrane at a given time in the experiment $t$ is assumed to be related linearly to the concentration of the QX species,

$$\frac{1}{\tau_F(t)} = \frac{1}{\tau_{F,0}} + k_{qE}([QV](t) + [QA](t) + [QZ](t)) + k_{qZ}[Z](t), \quad (1)$$

where $1/\tau_{F,0}$ is the intrinsic fluorescence decay rate of chlorophyll (arising from both the dominant non-radiative and minor radiative pathways), and $k_{qE}$ is the quenching rate constant for the QX complexes that mediate the LHCX1 and ΔpH-dependent qE quenching. We also incorporate zeaxanthin-dependent quenching, qZ, into the model by adding a quenching contribution that solely depends on the concentration of zeaxanthin in the "pool". The quenching rate constant for Z is denoted $k_{qZ}$. We assume that qE and qZ mechanisms are non-radiative, dissipating chlorophyll excitation energy as heat into the environment. From this we can obtain the experimentally measured $NPQ_\tau = (\tau_F(0) - \tau_F(t))/\tau_F(t)$. We assume that whilst the extent to which PX converts to QX under HL conditions is dependent on X, the quenching rate of each complex in the chloroplast is the same. With the available $NPQ_\tau$ data, we found that it is not possible to ascertain whether the differences in total quenching capacity of the different QX species arise due to differences in quenching rate, or the positions of the PX⇌QX equilibrium under HL conditions. Therefore, for simplicity, we treat the quenching rate $k_{qE}$ as being identical for all QX, and we also assume that the equilibrium constant for this process is zero in the dark.

The interconversion of the xanthophylls is assumed to occur after unbinding of X from P, PX⇌P + X. The X species in the model should be regarded as X in the pool of xanthophylls not bound to P. For example,

X could be bound to other light-harvesting proteins from which it can unbind rapidly and reversibly. The xanthophylls in the pool can be de-epoxidised sequentially, from V → A and then A → Z, by VDEa, where the maximum turnover rate for the VDE enzyme is different for the two de-epoxidation steps. VDE is assumed to interconvert between VDEa and VDEi forms depending on light conditions. We model this as a simple two-state equilibrium with first-order rate laws for the activation and deactivation. We also treat the epoxidation steps as sequential, first from Z → A then from A → V, and we assume that each epoxidation by the ZEP enzyme can be treated as a first-order rate process, with different epoxidation rates for Z and A.

### Dynamical response of xanthophyll concentrations to light stress

In order to investigate the response of the xanthophyll cycle to fluctuating light conditions, we have measured the changes in concentrations of these pigments in *N. oceanica* in response to four sequences of high-intensity light exposure: 5 HL- 10 D- 5 HL, 1 HL- 4 D- 7 HL- 5 D- 1 HL- 2 D, 10 HL- 10 D, and 1 HL-1 D, where HL denotes high light, D denotes darkness, and numbers indicate the duration of the exposure in minutes. The HPLC data showed a significant fraction of xanthophylls, particularly V, that remained constant over the time scale of the experiment, which we believe corresponds to xanthophylls strongly bound to proteins other than LHCX1. The samples were dark-acclimated for 30 min prior to HL exposure to ensure minimal initial amounts of A and Z. Figure 2 shows the change in VAZ cycle carotenoids relative to their initial dark-acclimated values (at $t = 0$), i.e. $\Delta[X] = [X](t) - [X](0)$ and $[X](t)$ is the total concentration of X at $t$. The experimental data show that $\Delta[A]$ was greater than $\Delta[Z]$ during HL exposures; $\Delta[A]$ remained relatively constant during dark periods (Fig. 2), which shows a more rapid dynamical response to reduction in light exposure. In the 5 HL- 10 D- 5 HL sequence (Fig. 2A), during the 10-minute dark period $\Delta[Z]$ decreased almost entirely back to its dark-acclimated value whilst $\Delta[A]$ remained constant for the first five minutes of darkness before it began to diminish. Both $\Delta[A]$ and $\Delta[Z]$ increased in response to the second HL exposure, and the rate of Z accumulation was greater than during the first HL period. Similarly in the 10 HL-10 D sequence (Fig. 2C), $\Delta[A]$ remained at a constant level compared to $\Delta[Z]$, which decreased more rapidly back to its dark-acclimated concentration. In the 5 HL- 10 D - 5 HL and 10 HL-10 D sequences, there was a small amount of continued accumulation of A and Z in the first dark phase for ~1 min, indicating a delayed deactivation of the de-epoxidation process, as we found previously in modeling the $NPQ_\tau$ response of *N. oceanica*[20].

Rate constants for xanthophyll interconversion in the model were parameterized based on a reduced form of the full model, fitted to the experimental HPLC data, as detailed in the Supplementary Information (Sec. 2). The full model predictions for the HPLC data are also shown in Fig. 2, where we see the model mostly predicts the HPLC data within the experimental fluctuations, although in the 1 HL- 4 D- 7 HL- 5 D- 1 HL- 2 D sequence the model slightly overestimates $\Delta[A]$ and $\Delta[Z]$ after 1 min of light exposure (it should be noted that the fluctuations in xanthophyll concentrations in Fig. 2D do not correlate with the periodicity of light exposure on close inspection). In Table 1 we summarize the maximum rates for the de-epoxidation processes, defined as $k_{X \to X'}$ [VDEa] $_{max}^{light/dark}$, and the epoxidation rates in the light and dark phases, and the rate constant for activation/deactivation (i.e. formation of VDEa from VDEi). We see that VDE activity increases by a factor of around 1000 in HL conditions, and that the VDE de-epoxidises A slightly faster than V, although the difference is small. Conversely for the epoxidation we see that Z is epoxidised nearly twice as fast as A. In our model, we find that the VDE enzyme takes just over 1 min to activate and deactivate in both the light and dark phases, which is consistent with the continuing increase in A and Z concentrations observed at the start of the dark phases in the HPLC experiments.

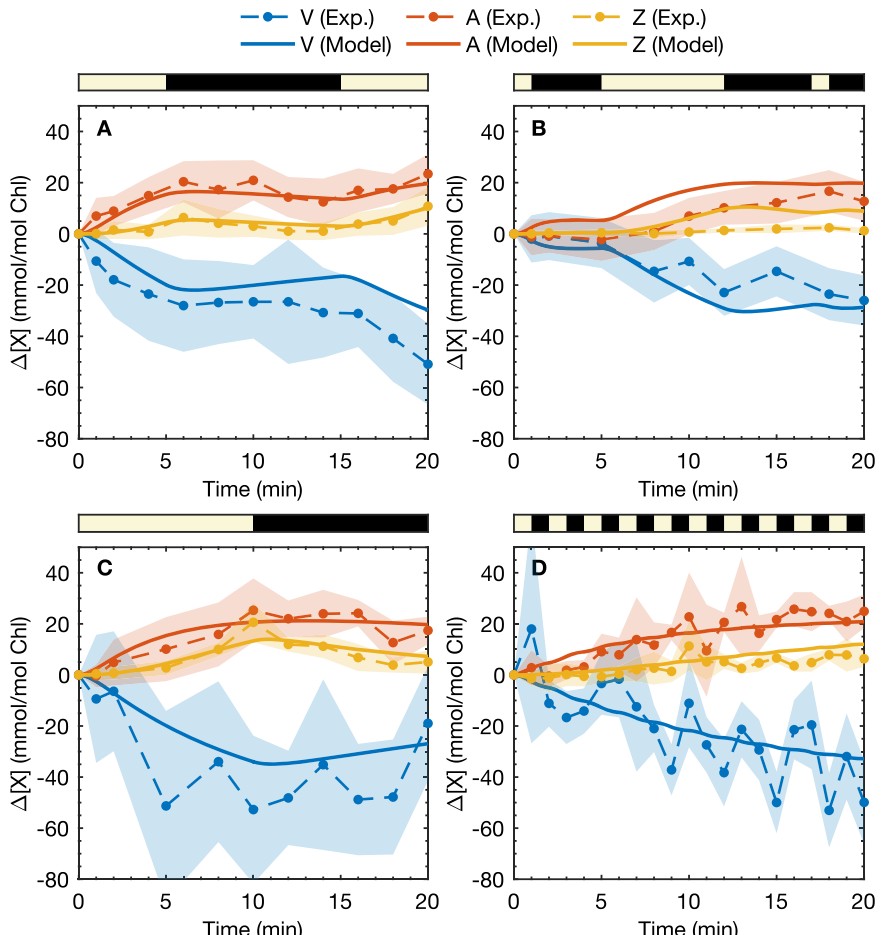

**Fig. 2 | Experimental HPLC data for the change in xanthophyll concentration.** $\Delta[X]$ as a function of time for four HL exposure sequences: **A** 5 HL- 10 D- 5 HL, **B** 1 HL- 4 D- 7 HL- 5 D-1 HL- 2 D, **C** 10 HL- 10 D, **D** 1 HL-1 D (yellow shaded regions indicate the HL phases). Experimental results are shown as points and model predictions are shown as solid lines. Predictions correspond to the total xanthophyll concentrations, $\Delta[X]_{tot} = \Delta[X] + \Delta[PX] + \Delta[QX]$. Experimental error bars (shaded regions) correspond to two standard errors of the mean (from $n = 3$ technical replicates). RMSD (root mean square deviations) in the fits are **A** $RMSD_V = 11.2$, $RMSD_A = 8.6$, $RMSD_Z = 11.5$ **B** $RMSD_V = 6.8$, $RMSD_A = 11.5$, $RMSD_Z = 3.1$ **C** $RMSD_V = 14.6$, $RMSD_A = 9.5$, $RMSD_Z = 8.4$, and **D** $RMSD_V = 11.7$, $RMSD_A = 11.6$, $RMSD_Z = 10.7$ all in mmol/ mol Chl $a$.

## Modeling NPQ response of *N. oceanica* to light exposure

Time-correlated single photon counting (TCSPC) experiments were also performed on *N. oceanica* to measure $NPQ_\tau$ in response to sequences of HL/dark exposure. In addition to 20-minute regular and irregular light sequences that were utilized in previous work[20], seven new HL/dark cycles were utilized to ascertain how long algae retain their "photoprotective memory" of previous HL exposure. The sequences had increasing dark durations ($T = 1, 5, 9, 10, 12, 15, 20$ min) between two five-minute HL periods. The model was employed to describe $NPQ_\tau$ dynamics of *N. oceanica* in response to various HL/dark exposure sequences, with parameters determined by fitting a subset of the $NPQ_\tau$ sequences, namely the 5 HL-9 D-5 HL, 5 HL-15 D-5 HL, 3 HL-1 D-1 HL-3 D-9 HL-3 D, 1 HL-2 D-7 HL-5 D-1 HL-2 D, 2 HL-2 D sequences [Fig. 3C, F, H, J]. Further details of this fitting procedure are given in the Methods section and Supplementary Information (Sec. 1).

The experimental $NPQ_\tau$ data are shown in Fig. 3. We see rapid NPQ activation and deactivation in response to changes in light levels, occurring on a timescale of <1 min, together with a slower increase in $NPQ_\tau$ during light exposure. The rapid component of $NPQ_\tau$ activation and deactivation arising from the pH-sensing protein corresponds to the equilibration rate for the PX equilibrium in the model, given by $k_{QX}^{light/dark} = k_{QX,f}^{light/dark} + k_{QX,b}^{light/dark}$. This equilibration rate is 2.1 min$^{-1}$ under light conditions and 4.7 min$^{-1}$ in the dark which gives an activation time

of 29 s and a deactivation time of 13 s. Experimental data for the 5 HL-$T$ D-5 HL sequences, Fig. 3A–G, show how $NPQ_\tau$ recovers after various dark durations, directly probing the photoprotective memory of *N. oceanica*. The $NPQ_\tau$ component recovered to its value at the end of the initial light period ($t = 5$ min) within 1 min upon secondary light exposure when dark durations were up to $T = 5$ min, and even with a 20 min dark duration, the $NPQ_\tau$ recovered within 3 min.

## Table 1 | Rate constants for xanthophyll interconversion steps for the full VAZ model

| Rate constant (min$^{-1}$) | HL conditions | Dark conditions |
|---|---|---|
| $k_{V \to A, max}$ | $0.092 \pm 0.02$ | $(9.1 \pm 6.2) \times 10^{-5}$ |
| $k_{A \to Z, max}$ | $0.14 \pm 0.05$ | $(1.4 \pm 1.0) \times 10^{-4}$ |
| $k_{Z \to A}$ | $(8.5 \pm 3.0) \times 10^{-2}$ | $(8.5 \pm 3.0) \times 10^{-2}$ |
| $k_{A \to V}$ | $(5.1 \pm 2.8) \times 10^{-2}$ | $(5.1 \pm 2.8) \times 10^{-2}$ |
| $k_{VDE}$ | $1.3 \pm 0.8$ | $1.0 \pm 0.75$ |

$k_{X \to X', max}$ is defined as $k_{X \to X'} [VDEa]_{max}^{light/dark}$, where $[VDEa]_{max}^{light/dark}$ is the maximum concentration of VDEa under light/dark conditions. $k_{VDE} = k_{VDE,f} + k_{VDE,b}$ is the rate constant for activation/deactivation of VDE, such that in a light/dark phase $[VDEa]$ changes according to $[VDEa](t) - [VDEa](t_0) = ([VDEa]_{max} - [VDEa](t_0)) e^{-k_{VDE}(t-t_0)}$. Errors given are two standard errors in the mean from bootstrapping.

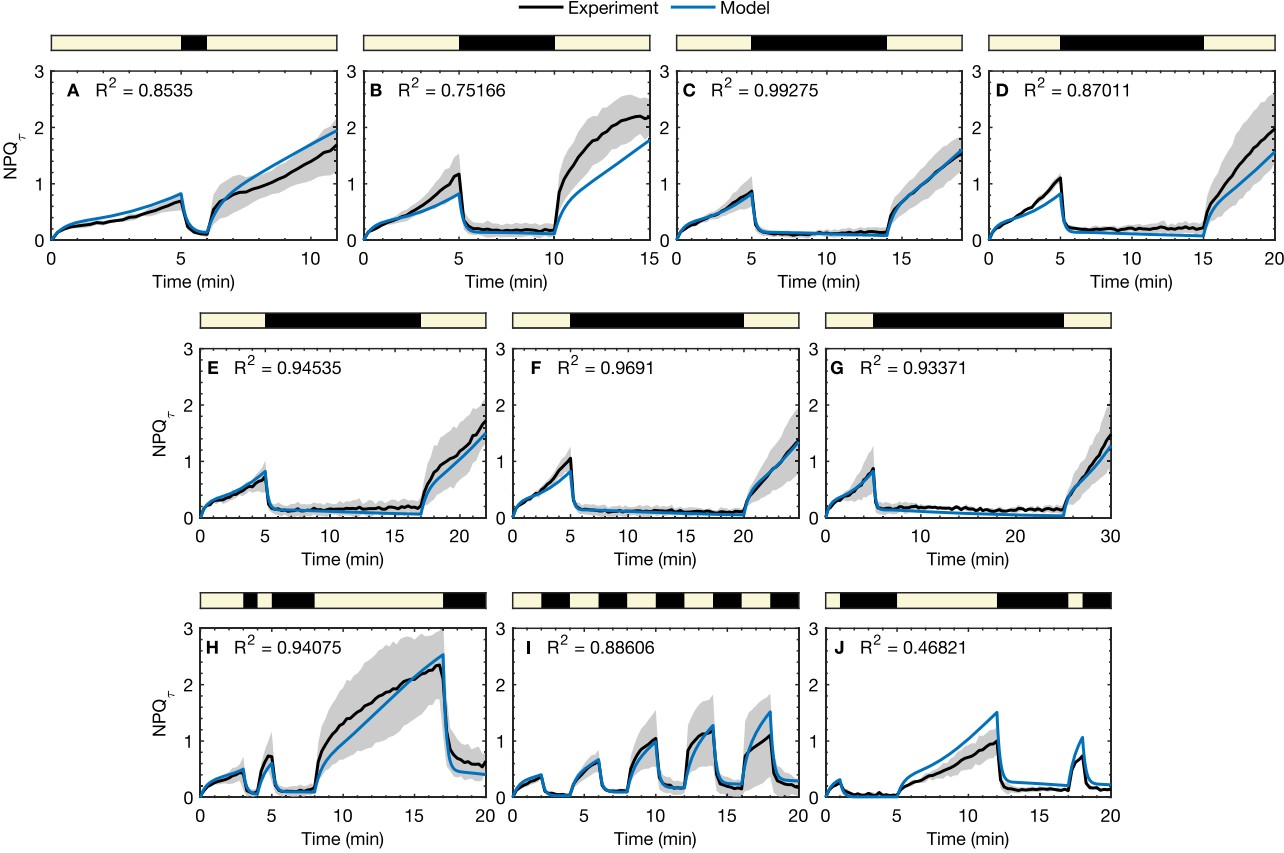

**Fig. 3 | Experimental NPQ$_\tau$ data (black) together with the model predictions for the NPQ$_\tau$ (blue) for various sequences of HL exposure/darkness for *N. oceanica*.** Yellow regions indicate HL phases of the experiments. Error bars correspond to two standard errors in the NPQ$_\tau$ measurements (from $n = 3$ technical replicates). **A–G** show data and model predictions for the 5 HL-$T$ D- 5 HL sequences and **H–J** show three additional sequences, where HL denotes HL exposure and D denotes darkness, with number indicating the exposure time in min. RMSD values for the fits are **A** 0.174 ($n = 3$), **B** 0.370 ($n = 3$), **C** 0.036 ($n = 3$), **D** 0.190 ($n = 3$), **E** 0.099 ($n = 3$), **F** 0.062 ($n = 3$), **G** 0.081 ($n = 3$), **H** 0.185 ($n = 3$), **I** 0.121 ($n = 3$), **J** 0.193 ($n = 3$).

In addition to the HPLC $\Delta[X]_{\text{tot}}$ data in Fig. 2, the model is able to predict the average NPQ$_\tau$ levels for all the sequences as shown in Fig. 3, including sequences other than those in the training datasets. Differences between the model predictions and experiments were generally comparable to the variability between experimental runs. For example at the end of the first five minutes of light exposure, NPQ$_\tau$ in the 5 HL- $T$ D- 5 HL sequences (Fig. 3A–G) the experimental NPQ$_\tau$ varies between around 0.8 and 1.4, although some discrepancies may be attributed to shortcomings of the model. Specifically the over-prediction of NPQ$_\tau$ for the 1 HL-2 D-7 HL-5 D-1 HL-2 D, 2 HL-2 D sequence [Fig. 3J] in the second light phase could be attributed to VDE activating too fast, as is seen in both the HPLC data and modeling [Fig. 2B].

In the model, the position of the PX⇌QX equilibrium under HL conditions determines how well each of the xanthophylls can act as a quencher in qE. The maximum fraction of P-bound X that can exist in the QX state under HL conditions, denoted $q_X$, determines the quenching capacity of each xanthophyll within our model. This can be expressed as

$$q_X = \frac{K_{QX}^{\text{light}}}{1 + K_{QX}^{\text{light}}} \tag{2}$$

where $K_{QX}^{\text{light}}$ is the equilibrium constant for the PX⇌QX process under HL conditions determined from fitting the model to the experimental data. In Table 2 we list these values for our model under light and dark conditions, obtained from fitting the model to the experimental NPQ$_\tau$ curves. From the $q_X$ values we find that A is approximately three times

more effective as a quencher than V, and Z is nearly 10 times more effective than V. From the model we can also quantify the relative contributions of qE and qZ to the overall quenching, by the ratio of $k_{qZ}$ to $k_{qE}$, which is found to be $k_{qZ}/k_{qE} = 0.026 \pm 0.005$.

## NPQ in *N. oceanica* mutants

To further test the model, we have modified the wild type (WT) *N. oceanica* parameterized model to predict the NPQ$_\tau$ response of two *N. oceanica* mutants: the *vde* and *lhcx1* mutants. The NPQ$_\tau$ response of the *vde* mutant, which has VDE knocked out preventing the accumulation of Z, was modeled utilizing parameters obtained from the WT model with $k_{V\to A}$ and $k_{A\to Z}$ to zero. The NPQ$_\tau$ response was measured for three HL/D sequences, shown in Fig. 4A–C together with model predictions. Even in the absence of A and Z, NPQ$_\tau$ increases near-instantaneously to around 0.3 in response to light, demonstrating the relevant role of LHCX1 in the *vde* mutant. However, because of V's low quenching capacity, the NPQ$_\tau$ response is significantly smaller than that seen in WT, and there is no steady increase of NPQ$_\tau$ over the duration of the experiment, unlike in the WT *N. oceanica*. The model captures the NPQ$_\tau$ response of the *vde* mutant remarkably well, despite not being parameterized with these data.

**Table 2 | Quenching capacity, $q_X$, for each of the xanthophylls**

| X | Violaxanthin | Antheraxanthin | Zeaxanthin |
|---|---|---|---|
| $q_X$ | 0.10 ± 0.02 | 0.28 ± 0.08 | 0.92 ± 0.05 |

Errors given are two standard errors in the mean.

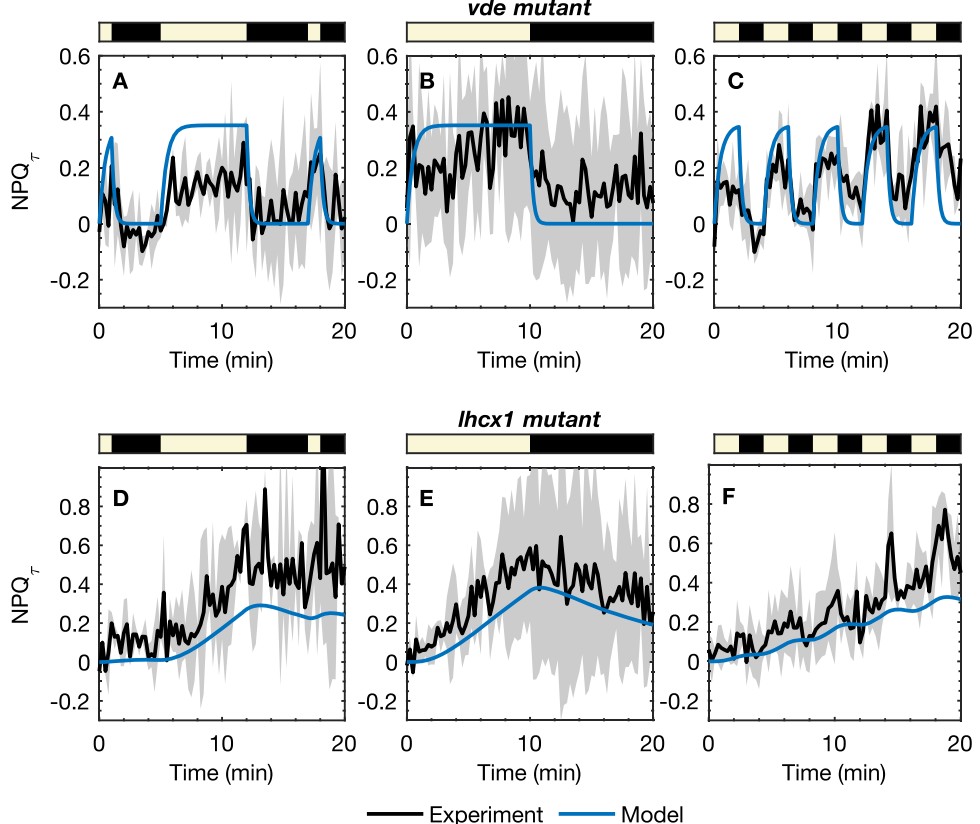

**Fig. 4 | NPQ$_\tau$ responses measured for the *vde* mutant (black) together with model predictions (blue) for three sequences of light/dark exposure.** Error bars correspond to two standard errors in the NPQ$_\tau$ measurements (from $n = 3$ technical replicates). Light/dark sequences: **A** 1 HL-2 D-7 HL-5 D-1 HL-2 D ($n = 2$), **B** 10 HL-10 D ($n = 2$), and **C** 2 HL-2 D ($n = 3$) × 5. **D**–**F** NPQ$_\tau$ responses were measured for the *lhcx1* mutant (black) together with model predictions (blue) for three sequences of light/dark exposure. Light/dark sequences: **A** 1 HL-2 D-7 HL-5 D-1 HL-2 D ($n = 2$), **B** 10 HL-10 D ($n = 3$), and C) 2 HL-2 ($n = 3$) D × 5. RMSD for the model predictions are **A** 0.134, **B** 0.136, **C** 0.118, **D** 0.227, **E** 0.139, **F** 0.142.

We have also modeled the NPQ$_\tau$ response of the *lhcx1* mutant, in which LHCX1 is not expressed and only zeaxanthin-mediated qZ quenching operates. This was modeled by simply setting [P]$_{tot} = 0$, removing the qE quenching process, while holding the total xanthophyll concentration constant. The experimental NPQ$_\tau$ data and model predictions are shown in Fig. 4D–F, where we see the model accurately captures the slow rise of NPQ$_\tau$ in the light phases, arising from the build-up of Z during light exposure, and the slower decay in the dark phases due to slow epoxidation of Z. The success of the model in predicting the NPQ response of the *lhcx1* mutant strongly supports the interpretation of the kinetic model species "P" as involving or at least requiring LHCX1 to function.

## Discussion

Our combined experimental and kinetic model results suggest that photoprotective memory in *N. oceanica* can be explained qualitatively with a simple three-state model. The three-state system should consist of a poor quencher (V), a modest quencher (A), and a good quencher (Z). After a sample has sufficiently accumulated the good quencher, during brief dark/low-light periods, Z remains before being converted back to the modest quencher (A), acting as short-term memory. However, during extended dark durations, Z will be converted almost entirely to A. Whilst A is also converted back to V, the A → V transition occurs at a slower rate such that during another HL exposure occurs, the Z pool can form more rapidly from the reservoir of A. We can also see this dynamic represented in the HPLC data (Fig. 2). By adding the intermediate step in the VAZ cycle, the model not only more accurately reflects the biochemical processes but also allows for the short-term

photoprotective memory, over time scales between 1 min to ~30 min, to be modeled and understood.

From our experiments and modeling, we have also been able to determine the relative quenching capacities of the different xanthophylls. We find that V facilitates a weak but rapid response to changes in HL. The *vde* mutant demonstrates that even without an effective quencher like Z, there is still an NPQ$_\tau$ response to fluctuating light. In very short bursts of HL, V may act as the main quencher where the switch between its roles in photochemistry and photoprotection is determined by the pH gradient, as suggested previously[32].

As the intermediate step in the VAZ cycle, A's role as a potential quencher in qE is often overlooked. With a quenching capacity of around 30%, it is 3.5 times less efficient than Z (95%) at dissipating excess energy. However, it plays a crucial role in photoprotection in facilitating NPQ recovery after long dark durations. In Fig. 5 we show a breakdown of the NPQ$_\tau$ response predicted by the model for the 5 HL-10 D-5 HL sequence, where we see at short times the main quencher in qE is actually V complexed with LHCX1, with contributions from A emerging at $t = 1$ min and Z at $t = 2$ min. After light exposures of more than 2 min, Z functions as the primary quencher, with small, but not insignificant, contributions from V and A. Whilst LHCX1-dependent qE makes the largest contribution to NPQ$_\tau$, qZ also makes a small contribution, and within the model, this is the sole contributor to the long-lived NPQ$_\tau$ response in the dark. Even for very long-time light exposure, the model predicts that LHCX1-dependent qE dominates over qZ, with qZ making up only ~23% of the total NPQ$_\tau$ in this limit, whilst the LHCX1-Zeaxanthin qE accounts for the majority (~75%) of the limiting NPQ$_\tau$. It should be noted that this limit is based on extrapolating the

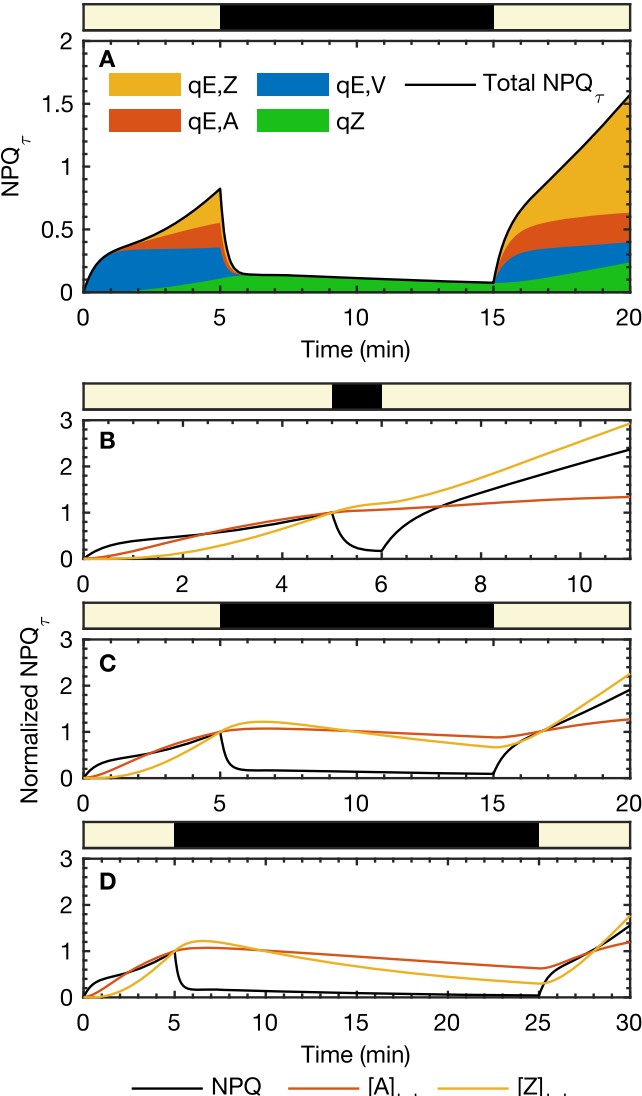

**Fig. 5 | Contributions of each xanthophyll to the total NPQ$_\tau$. A** Contributions of each xanthophyll to the total NPQ$_\tau$ as predicted by the model as a function of time for the 5 HL- 10 D-5 HL sequence. **B–D** NPQ$_\tau$, [A]$_{tot}$ = [A] + [PA] + [QA], and [Z]$_{tot}$ = [Z] + [PZ] + [QZ], predicted by the model for three 5 HL-$T$ D-5 HL sequences of light/dark exposure: **B** $T$ = 1 min, **C** $T$ = 10 min and **D** $T$ = 20 min.

model to light exposure times beyond those that we have investigated, which may not be reliable, and we also expect the relative contributions of qE and qZ to depend strongly on species and growth conditions, as has been found in studies of plants[22,33,34]. We have not suggested a microscopic model for the qZ process, although in the SI, Sec. 4, we show how a quenching process depending on some other zeaxanthin binding protein (or protein complex) P′ would be consistent with our simple model. Zeaxanthin binding to some other protein could activate qZ by directly quenching excitation energy, potentially via charge transfer, or inducing conformational changes in the protein that promote other quenching mechanisms[35–37].

An essential element of the three-state photoprotective memory system observed in *N. oceanica* is the kinetics of xanthophyll cycle, which together with the quenching capacities of the xanthophylls creates an effective photoprotective system. Upon the first exposure to light, NPQ activation is limited by moving through two steps before Z, the primary quencher, is accumulated, where VDE activation and the V → A step (with a half-life of ~7 min) control the initial rate of NPQ activation. Z may still function as a moderate quencher in the dark

through qZ, so fast conversion of Z → A by ZEP (half-life ~8 min) in the dark is necessary to facilitate efficient photosynthesis under low-light conditions. The slower kinetics of A → V in the dark (with a half-life ~20 min) enables A to function as a buffer, facilitating rapid NPQ reactivation if light levels fluctuate again to damaging levels. The fast A → Z conversion by VDE on light exposure (with a half-life of ~4 min) also plays an essential role in photoprotective memory by enabling the buffer of A to be rapidly converted to an active quencher. Previous work in plants found the rate of de-epoxidation of A to be about 4 times faster than that of V[29–31], which is a much larger difference compared to the de-epoxidation rates that we have found, with de-epoxidation of A being only about 1.5 times faster than that of V. However, VDE activity is influenced by the thylakoid lumen acidity, availability of ascorbate, and potentially unique species-specific differences, any of which could explain this discrepancy. Furthermore, because VDE is not active in the dark, the relative activity of ZEP on Z and A is far more relevant to photoprotective memory than the relative activity of VDE on V and A. On top of the slower time scale kinetics of the VAZ cycle, which control the maximum quenching capacity of the system, very rapid responses to light fluctuations, on time scales of around 1 min or less, are facilitated by protonation and subsequent conformational changes of the quenching protein which binds the xanthophylls.

From the model, we can directly probe how the total A and Z concentrations vary during the 5 HL-$T$ D- 5 HL sequences to demonstrate the functional role of xanthophyll cycle kinetics in photoprotective memory. Here we show in Fig. 5 the model NPQ$_\tau$ and the total A and Z concentrations normalized by their values at $t$ = 5 min. For very short dark phase ($T$ = 1 min, Fig. 5B) Z continues to accumulate (due to the finite deactivation time of VDE in our model), acting as short-term light exposure memory and the NPQ$_\tau$ recovers very rapidly upon re-illumination. For intermediate and longer lengths of dark duration ($T$ = 10 min, Fig. 5C and $T$ = 20 min, Fig. 5D), the quencher Z decreases but A remains steady, presumably acting as a buffer, and thus as a short-term memory for excess light exposure, and facilitating a fast response to HL in the second light phase. In these cases, the NPQ$_\tau$ response in the second HL phase correlates most strongly with the A concentration, and not the Z concentration. In the Supplementary Information, Fig. S2, we show the experimental and model NPQ$_\tau$ recovery, averaged over the first minute of HL, in the second light phase for the 5 HL-$T$ D- 5 HL sequences, as a function of dark duration $T$. From this we have extracted (see Supplementary Information Sec. 2 for details) an NPQ$_\tau$ memory time scale of ~22 min, which matches the model A → V time scale given by $1/k_{A\to V}$ = 19.9 min. This strongly suggests that antheraxanthin acts as a short-term memory for light exposure, with the A → V step of the xanthophyll cycle controlling the effective memory time scale. It has previously been observed that xanthophyll composition correlates with photoprotection, long- and medium-term light-exposure memory, and light levels during growth in plants[23,34], phytoplankton[24,25] and algae[26]. We can now however add to this picture that the kinetics of the xanthophyll cycle also plays an important role in short-term photoprotective memory.

One important quantity we can estimate from this study is the lifetime of Chl-*a* excitations on the active quenching complexes QX. Firstly from the HPLC data and model we obtain an estimate of the total concentration of P (possibly LHCX1 or LHCX1 in a complex with other proteins) in the system as ~0.6 mmol/mol Chl. Assuming roughly ten Chl-*a* molecules per light-harvesting protein, this means the species P makes up ~1 in 30 light-harvesting proteins in *N. oceanica*. Using this ratio of P to the other light-harvesting proteins and assuming excitation energy diffusion between proteins is faster than quenching, we can estimate the lifetime of Chl-*a** on the active quenchers to be less than ~10 ps (further details of this calculation are given in the SI, Sec. 4). This approximate time scale is roughly consistent the quenching time scale in HL acclimated *N. oceanica* observed in

transient-absorption experiments of ~8 ps[4] (especially given the simplifying assumptions we use to deduce our estimate). Recent work has suggested that quenching can be limited by excitation energy redistribution within and between light-harvesting proteins[36,38,39], so the actual quenching process (likely either excitation energy transfer or charge transfer quenching[4]) may need to occur on an even shorter time scale than this estimate.

Overall in this work, we have presented a model of xanthophyll cycle mediated non-photochemical quenching in *N. oceanica*, which can both accurately describe the short and intermediate timescale NPQ$_\tau$ responses of *N. oceanica* to HL stress and the accompanying changes in xanthophyll concentrations. Employing a combination of experiments and modeling we have developed a deeper understanding of the photoprotective roles of the xanthophylls together with LHCX1. From this, we have suggested a three-state model for short time scale photoprotection in *N. oceanica*, where the zeaxanthin-LHCX1 system acts as the primary quencher, with antheraxanthin acting as a short-term "memory" of HL stress capable of facilitating rapid response to fluctuations in light levels, and violaxanthin deactivating quenching under low-light conditions. This adds to the established picture of xanthophyll composition correlating with long-term memory of light-exposure[22]. Although we cannot conclusively identify the qE quencher, PX/QX, we can say that LHCX1 is an essential component of this system. We have also been able to estimate the chlorophyll excitation lifetime on active quenching proteins as less than ~10 ps, as well as the relative abundance of quenchers in the thylakoid membrane. Evidence for zeaxanthin-dependent but LHCX1-independent "qZ" quenching has also been found, although its contribution to NPQ appears to be much smaller than that of LHCX1-dependent "qE" quenching. However, the proportion of qE or qZ contributions is going to vary depending on the species[40]. In order to implement a similar model of NPQ for use in vascular plants, more components need to be incorporated such as quenching due to lutein and state transitions[5–7], which are not present in *N. oceanica*. However, we believe the model presented here provides a basis for building a quantitative model of NPQ responses for plants and other photosynthetic organisms, which are mediated by the same xanthophyll cycle.

## Methods

### Algal growth conditions
*N. oceanica* CCMP1779[6] was obtained from the National Center for Marine Algae and Microbiota (https://ncma.bigelow.org/) and cultivated in F2N medium[41]. Liquid cultures were grown to $2–5 \times 10^7$ cells/mL in continuous light at a photon flux density of 60 μmol photons m$^{-2}$ s$^{-1}$ at 22 °C or room temperature.

The knock-out mutants *vde* and *lhcx1* (Ref. 4) were generated using homologous recombination of a hygromycin resistance cassette, with the addition of Cas9 RNP for *lhcx1*. Further details of how the mutants were generated will be presented in a separate manuscript.

### Time-correlated single photon counting
Time-correlated single photon counting results in a histogram of Chl-*a* fluorescence decay, which is then fit to a biexponential decay function yielding an average lifetime ($\bar{\tau}$). Fluorescence lifetime measurements were taken every 15 seconds to capture the change in fluorescence lifetimes as a function of HL exposure. The amplitude-weighted average lifetime of the Chl-*a* fluorescence decay is converted into a unitless form, similar to that measured in the conventional pulse-amplitude modulation technique using the following equation: where $\bar{\tau}(0)$ and $\bar{\tau}(t)$ are the average lifetimes in the dark and at any time point $t$, respectively, during the experiment.

$$\text{NPQ}_\tau = \frac{(\bar{\tau}(0) - \bar{\tau}(t))}{\bar{\tau}(t)} \qquad (3)$$

An ultrafast Ti:sapphire coherent Mira 900 oscillator was pumped using a diode laser (Coherent Verdi G10, 532 nm). The center wavelength of the oscillator was 808 nm with a full width at half maximum of 9 nm. After frequency doubling the wavelength to 404 nm with a $\beta$-barium borate crystal, the beam was split between the sample and a sync photodiode, which was used as a reference for snapshot measurements. Three synchronized shutters controlled the exposure of actinic light and the laser to the sample as well as to the microchannel plate-photomultiplier tube detector (Hamamatsu106 R3809U). The shutters were controlled by a LABVIEW software sequence. The detector was set to 680 nm to measure Chl-*a* emission. During each snapshot, the laser and detection shutters were opened, allowing an excitation pulse with a power of 1.7 mW to saturate the reaction center for 1 second while the emission was recorded. During HL periods, samples were exposed to white light with an intensity of 885 μmol photons m$^{-2}$ s$^{-1}$ (Leica KL 1500 LCD, peak 648 nm, FWHM 220 nm) by opening the actinic light shutter. The *N. oceanica* sample was concentrated at 40 μg Chl mL$^{-1}$. To do this, 1 mL of *N. oceanica* culture was pelleted for 5 minutes at room temperature at max speed, flash frozen in liquid nitrogen, thawed at room temperature, and broken using FastPrep-24 (MP Biomedicals LLC) at 6.5 m/s for 60 seconds. The pellet was flash-frozen and broken two more times. Chlorophyll was extracted from the broken cells using 1 mL of 80% acetone, and total chlorophyll in the culture was quantified according to Porra et al.[42]. The culture was then concentrated by centrifuging for 5 minutes at room temperature at 3320 g. Samples were dark-acclimated for 30 minutes prior to the experiment and placed in the custom-built sample holder on a sample stage. The LABVIEW sequence was altered for each regular, irregular, and dark duration sequence run to control exposure to light fluctuations. For the NPQ$_\tau$ experiments, three technical replicates were performed for the WT and three for each mutant. Two experimental replicates were performed for the 5 HL-*T* D-5 HL experiments and the training data for the model. Only one experimental replicate was performed for the mutants.

### High-performance liquid chromatography
Aliquots of *N. oceanica* in F2N media were taken at various time points during several regular and irregular HL/dark duration actinic light sequences. Samples were then flash-frozen in liquid nitrogen. After thawing, the samples were pelleted for 5 minutes at 4°C at maximum speed to reach a cell count of ~45–60 × 10⁶. The cells were washed twice with dH₂O and pelleted at maximum speed for 5 minutes. The cells were again flash-frozen and thawed at room temperature followed by breaking the cells using FastPrep-24 (MP Biomedicals LLC) at 6.5 m/s for 60 seconds. The bead beating step was repeated once before adding 200 μL of 100% cold acetone. The samples were centrifuged for 10 minutes (maximum speed, 4°C), and the supernatant was filtered (0.2 μm nylon filter) into HPLC vials. The supernatant was separated on a Spherisorb S5 ODS1 4.6- × 250 mm cartridge column (Waters, Milford, MA) at 30°C. Analysis was completed using a modification of García-Plazaola and Becerril[43]. Pigments were extracted with a linear gradient from 14% solvent A (0.1M Tris-HCl pH 8.0 ddH20), 84% (v/v) solvent B (acetonitrile), 2.0% solvent C (methanol) for 15 minutes, to 68% solvent C and 32% solvent D (ethyl acetate) for 33 min, and then to 14% solvent A (0.1M Tris-HCl pH 8.0 ddH2O), 84% (v/v) solvent B (acetonitrile), 2.0% solvent C (methanol) for 19 min. The solvent flow rate was 1.2 mL min$^{-1}$. Pigments were detected by A445 with reference at 550 nm by a diode array detector. Standard curves were prepared from isolated pigments. The HPLC peaks were normalized to the total Chl-*a* concentration.

### Model details
Each step of the model given in Fig. 1 is treated as an elementary reaction step in the 12 species model. As described in our previous work[20], we cannot determine from these experiments the absolute

concentration of VDE, so we replace the VDE species with a dynamical quantity $\alpha_{VDE}(t)$ representing the activity of VDE at a time $t$ relative to its maximum possible value. We also work in a reduced unit system defined for species B by $[\widetilde{B}] = \tau_F(0)k_{qE}[B]$, where $\tau_F(0)$ is the fluorescence lifetime at $t = 0$. With these reduced variables $NPQ_\tau(t) = \Delta[\widetilde{QV}](t) + \Delta[\widetilde{QA}](t) + \Delta[\widetilde{QZ}](t) + (k_{qZ}/k_{qE})\Delta[\widetilde{Z}](t)$, where $\Delta[\widetilde{QX}](t)$ is the change in reduced concentration of QX relative to the $t = 0$ value, and likewise for $\Delta[\widetilde{Z}](t)$.

The model parameters were fitted by minimizing the sum of square differences between the model $NPQ_\tau$ and the experimental $NPQ_\tau$ for the 5 HL-9 D-5 HL, 5 HL-15 D-5 HL, 3 HL-1 D-1 HL-3 D-9 HL-3 D, 1 HL-2 D-7 HL-5 D-1 HL-2 D, 2 HL-2 D sequences. Parameters for the VAZ interconversion steps were estimated from a least squares fit of a reduced model, which is a simple first-order kinetic model with activation of the VDE enzyme, to the HPLC data (this is detailed in the SI). In the rest of the parameter fitting these parameters were constrained to only vary by 50%. Additionally, to reduce the number of free parameters, the forward and backward binding rate constants $k_{PX,f}$ and $k_{PX,b}$, and the activation rate to form QX, $k_{QX}^{light/dark} = k_{QX,f}^{light/dark} + k_{QX,b}^{light/dark}$, were set to be independent of the species X. This way the equilibrium constant $K_{QX}$ is the only parameter in the model controlling the quenching capacity of each xanthophyll. The remaining parameters were fitted first using Matlab's "globalsearch" function from an initial guess based on our previous model. This was then refined using the "patternsearch" algorithm. Errors in the fitted parameters were estimated by bootstrapping the experimental time series 1000 times. The conversion factor from reduced units to the mmol/mol Chl units the HPLC data are reported in was found using a simple least squares fit between the experimental HPLC and model $\Delta[X]_{tot}$ values shown in Fig. 2. Full details of the model kinetic equations and the full parameter set are given in the SI.

## Reporting summary

Further information on research design is available in the Nature Portfolio Reporting Summary linked to this article.

## Data availability

All data presented in this manuscript is available at https://doi.org/10.5281/zenodo.8284422. Source data are provided with this paper.

## Code availability

All Matlab code used to run the model and produce figures in this manuscript is available at https://doi.org/10.5281/zenodo.8284422.

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

## Acknowledgements
The experimental work was supported by the U.S. Department of Energy, Office of Science, Chemical Sciences, Geosciences, and Biosciences Division through FWP 449B to K.K.N. and G.R.F. T.P.F. and D.T.L. were supported by the US Department of Energy, Office of Science, Basic Energy Sciences, CPIMS Program Early Career Research Program under Award DE-FOA0002019. K.K.N. is an investigator of the Howard Hughes Medical Institute. D.T.L. acknowledges support from the Alfred P. Sloan Foundation.

## Author contributions
G.R.F. and A.S. conceived the research. A.S. performed all spectroscopic and HPLC experiments, and performed initial data analysis. T.P.F. developed and implemented the model and performed the final data analysis. T.C. prepared all algal samples and generated mutants. R.M. assisted A.S. in performing experiments. A.S., T.P.F., and G.R.F. wrote the manuscript. A.S., T.P.F., T.C., G.R.F., D.T.L., and K.K.N. discussed the results and commented on the manuscript. D.T.L., K.K.N., and G.R.F. procured funding.

## Competing interests
The authors declare no competing interests.
