## [Peer Review File · Nature Communications]

Kinetics of the Xanthophyll Cycle and its Role in the Photoprotective Memory and ResponseReviewer #1 (Remarks to the Author):

-This work makes a significant novel contribution to the field. Noteworthy results include important and novel contributions to the understanding of photosynthetic organisms' photoprotection against damaging excess light. This study uses the elegant system of an organism possessing the widespread xanthophyll-cycle system used by many photosynthetic organisms but without several added complications. Complications absent in this attractive model system include the additional xanthophyll lutein as well as additional processes (state transitions) with confounding effects on the non-photochemical fluorescence quenching critical to the analysis of photoprotective energy dissipation of excess energy. This study system allows valuable in-vivo characterization of photochemical processes without the need to resort to using isolated system with potential compromised responses.

This manuscript specifically focuses on the xanthophyll-cycle intermediate antheraxanthin that has been elusive and difficult to study. The manuscript reveals what promises to be an important function of antheraxanthin as a memory for prior exposure to excess light and an organism's ability to fully protect itself in rapidly fluctuating light environments that are so common in nature.

-The work supports the conclusions and claims made.

-The methodology is sound, although some of the specific details are not the primary area of my expertise. Enough detail is presented to allow the work to be reproduced.

Specific comments

-Note that non-photochemical quenching per se dissipates only a very small fraction of excess excitation energy. The major pathway of non-photochemical energy utilization leads to thermal energy and not to fluorescence emission. Avoid implying that non-photochemical fluorescence quenching is a major pathway of energy dissipation.

-Consider defining HL as high light upon first mention and then consistently using the abbreviation for the rest of the text.

-The authors might consider elaborating on what had previously been reported on the topic of some sort of plant memory for excess light (examples of reviews touching on this subject include *Plant Physiol. Rep.* 2022, 27:560–572; *Molecules* 2020, 25, 5825).

-Concerning the elusive role of antheraxanthin in photoprotective energy dissipation, the authors could mention that correlations between chlorophyll fluorescence parameters and pigment levels reported for plants were higher when including A in addition to Z (see, e.g., *Plants* 1996, 198, 460-470).

-Where they comment on the relative contributions of qE and qZ in *N. oceanica*, the authors might consider mentioning that these features are impacted by species and growth light environment (different plant species, as well as plants of the same species grown in different light environments, vary widely in their maximal NPQ and in the proportions of qE and qZ; see, e.g., Figure 2 in *Plant Physiol. Rep.* 2022, 27:560–572; *Photosynthesis Research* 2012, 113, 75-88; *Planta* 1998, 205, 367-374).

Reviewer #2 (Remarks to the Author):

The manuscript "Kinetics of the Xanthophyll Cycle and its Role in the Photoprotective Memory and Response" presents a model and experimental data for how short term photoprotective memory works in the microalgae *Nannochloropsis oceanica*. I believe this is both novel and an exiting interpretation of how each part of the xanthophyll cycle impacts NPQ.

The work is original and will help researchers further dissect the contribution and kinetics of individual mechanisms that collectively result in the NPQ response. The way the xanthophyll's

contribution to qE was resolved in this work is particularly interesting.

Concern that should be addressed before publication:

1) The authors state "We see rapid NPQ activation and deactivation in response to changes in light levels, occurring on a time-scale of less than 1 min, together with a slower increase in NPQ τ during light exposure." and this appears to be correct however, a quantification of activation/deactivation time would be good to be included here (e.g. decay half time).

This will also help support their next argument "We see the NPQ τ recovers to its value at the end of the initial light period ($t = 5$ min) within 1 min in the second light period for dark durations up to $T = 5$ min, and even with a 20 min dark duration the NPQ τ recovers within 3 min.". Please provide some quantification of the NPQ activation time, or show those recovery points on the figures. A single exponential decay fitting works well on PAM derived NPQ, I expect that it would work with NPQt, see the following paper for such an example: <https://doi.org/10.1038/s41598-019-44972-z>

2) In Figure 3 A to G the only interesting part to visualise is the second illumination phase, which would be most interesting to see when all curves are combined in to one single plot. That way we can better assess the effect on NPQ induction of the different dark intervals. I suggest either modifying Figure 3 or to provide a supplementary figure where the above comparison is made.

3) In Figure 4 NPQt reaches negative values in multiple instances – please explain in the text why that is.

4) I think some statistical evaluation of how well the model predicts the experimental data is needed (e.g. residual analysis, Chi-Square goodness-of-fit test, or something similar). I believe that would be needed for each individual figure where the model and experimental data are compared, including the figures with the training data.

5) Please mention in the text the number of technical (if any) and biological replicates, as well as the number of experimental replication (how many times was this work repeated)?

6) Standard deviation is preferred to the standard error of mean

Minor observations:

1) This needs rephrasing "The accumulation of A and Z is correlated with increases in NPQ throughout a diurnal cycle in plants,^{10,11} and it has also been proposed that slower activating and de-activating quenching mediated by Z in the absence of a pH-gradient sensing protein, termed "qZ", also operates in photosynthetic organisms.^{12,13}"; qZ is not a pH sensing protein

2) Make sure *N. oceanica* is in italic throughout the manuscript

3) A few misspellings that need correction: "occurring", "disprepancies"

4) This is just an aesthetic suggestion: the white and yellow colouring that represent dark and light it's not ideal. The black and white bar system used in your previous paper is easier to follow (<https://doi.org/10.1063/5.0089335>)

Overall, I believe the manuscript presents novel and interesting findings, with significant implications for understanding the photoprotective memory in *Nannochloropsis oceanica*. Once the authors address the concerns outlined above, particularly with respect to providing statistical validation of the model predictions and more detailed quantification of the NPQ response time, I anticipate that this work will be a valuable contribution to the field. I look forward to seeing the revised manuscript.

Andrei Herdean

Reviewer #3 (Remarks to the Author):

The manuscript deals with deciphering and modeling the kinetics of photoprotective xanthophyll pigments cycling (XC) and non-photochemical quenching (NPQ) in the microalga *Nannochloropsis*. The manuscript is well written, and the experimental data and modeling are convincing and of general value, i.e. XC and NPQ are photophysiological processes crucial to the photosynthetic productivity of most of all photosynthetic organisms, uni- and multi-cellular, aquatic and terrestrial. Nevertheless, there are, to my feeling, some lacks and unclear/uncomplete statements that require additional work and revisions before the manuscript could be published in *Nature Communications*. For these reasons, I suggest 'major revisions'. More detailed comments below.

Major comments:

-The nucleus of the model is based on the fact that a protein P binds a xanthophyll pigment (PX complex) and converts into a quencher complex (QX). This is fine. What is not is to consider that P is LHCX1 when we still do not know if LHCX1 (and any other LHCX protein) can effectively binds xanthophyll pigments, even more de-epoxidised forms, this questioning is even stated by the authors themselves in their introduction 'and whether or not LHCX1 can bind pigments is still under investigation'. Therefore, there is a central point I do not understand here, and which needs clarification. Furthermore, and based on this fact, I do not understand how (Part A-Results) 'the new model can quantitatively describe xanthophyll concentrations in cells, enabling us to estimate the absolute abundance of LHCX1': it would mean that it was demonstrated that LHCX1 effectively binds Zeaxanthin (Zx), which is not the case (or did I miss a recent report?). There are some strong indications but no more so far (see ref 16 and 19 here).

-All along the manuscript, starting with the title, the authors claim about a 'photoprotective memory'; how the authors came to this statement is unclear and the vocabulary used is itself obviously unclear to them: title and elsewhere in the manuscript (photoprotective memory), abstract (kind of memory, medium-term 'memory'), part C of the Results ('photoprotective memory'), etc. If the authors want to make this point the nucleus of their story (which I am not convinced it is), they should chose a wording and better explain this process. Also, if the authors want to strengthen this aspect, they should generalize to other organisms (see for instance <https://doi.org/10.3354/meps12398>) and to the in situ situation where the XC kinetics have been used to trace the water masses vertical mixing (numerous works by Brunet et al and by Bidigare et al, see for instance [doi:10.1093/plankt/fbu069](https://doi.org/10.1093/plankt/fbu069) ; [doi:10.1093/plankt/fbr102](https://doi.org/10.1093/plankt/fbr102) and references therein). This 'memory' aspect of the XC is not new and the authors should make better use of what has been published (even if not always the same time range) to strengthen their proposal.

-Experiments: as much as the modeling versus experimental data is generally convincing (see minor comments below), I feel there are some obvious experiments that could be easily added to challenge the modeling procedure and, I am sure, strengthen and even more generalize the output of this work. I especially see two kinds of experiments (based on the HL-D sequences performed by the authors) : 1) the same kind of HL-D sequences but with sustained Zx (i.e. $qZ \neq 0$ at the beginning of the experiment), which could be, I guess, obtained by growing *Nannochloropsis* under a higher light intensity; 2) the same kind of HL-D sequences but replacing D phases by Low Light in order to speed up Zx epoxidation and NPQ relaxation (works by Goss et al, ref 11 here). $qZ \neq 0$ is especially paramount as more and more works report that in in situ conditions qZ is rarely 0 and there is often, if not always, sustained Zx and other de-epoxidized xanthophylls.

Minor comments:

Abstract:

- 'quantitative model we show': dot missing between model and we show.

- last sentence: why only vascular plants? There is a growing blue biotechnology field, mostly based on marine microalgae (among which *Nannochloropsis*) that would be interested in such a work.

Introduction:

- *N. oceanica* in italics.

- 'a pH-sensing protein LHCX1': this is contrary to current knowledge and to the first sentence/2nd column of the abstract-page 1 (statement 'still under investigation'), indeed we do not know yet what exactly is the role of LHCX1, and there are recent works even suggesting it does not sense

pH at all (see ref 16 and 19 here).

-‘more complex features’: I do not understand why state-transitions are a ‘more complex feature’ than the XC and NPQ, this is wrong.

Results:

-HPLC data-Figure 2: units, is ‘Chl’ Chlorophyll a ? Because these data are normalized over Chl a (I guess), it is needed to show that during all experiments the Chl a content/cell does not change; regarding the length of the experiments, I guess there is no change but still this is important to confirm.

-page 4, column 1, top: ‘the model accurately predicts’, no, this is an overstatement that needs to be modulated; the model works fine to predict a trend, especially when the light climate is rather stable (Figure 2) but it does not work so well in ‘accurately’ predicting changes in XC dynamics when light fluctuates (Figure 2 D).

-Part D-Results: there is no information on the mutants: we can guess these are KO mutants, but otherwise we do not know how they were produced, especially from this work or from another piece of work, and if from a previous work, there is no reference.

-Part D-Results, last paragraph: KO LHCX1 mutant, there is no explanation (in the corresponding Discussion) on how NPQ induction is feasible in such mutant; see ref 15 here.

Discussion:

-bottom page 5: ‘poor quencher, modest quencher and good quencher’, and just below A is not anymore depicted as a ‘modest’ quencher but as an ‘intermediate’ one; these qualitative subjective statements should be avoided.

Conclusion:

-‘qZ contribution to NPQ being much smaller than the one of qE’: no, it depends on species and on environmental conditions; see the works by Demmig-Adams et al., Verhoeven et al and Lacour et al. on sustained qZ-like NPQ in tropical and desert plants, evergreen conifers and polar diatoms.

MandM:

-Part C-HPLC: ‘Standard curves from concentrated pigments’, I do not understand what is meant by ‘concentrated pigments’ ?

List of references:

-species names in italics.

-some references need to be updated (for ex. 14).

Below we address review comments on NCOMMS 23-25582 “The Xanthophyll Cycle and its Role in the Photoprotective Memory and Response” By Short, Fay et al.

We thank all the reviewers for their careful reading and appreciate their comments on the originality and usefulness of our work. We also appreciate the additional breadth in the context of our work provided by reviewers 1 and 3 which we have added to the text. We would particularly like reviewer 1 for the detailed reading of the manuscript and suggestions for improvements.

We have formatted the manuscript according to the Nature Communications instructions.

Taking the specific comments in detail:

Reviewer #1 (Remarks to the Author):

-This work makes a significant novel contribution to the field. Noteworthy results include important and novel contributions to the understanding of photosynthetic organisms' photoprotection against damaging excess light. This study uses the elegant system of an organism possessing the widespread xanthophyll-cycle system used by many photosynthetic organisms but without several added complications. Complications absent in this attractive model system include the additional xanthophyll lutein as well as additional processes (state transitions) with confounding effects on the non-photochemical fluorescence quenching critical to the analysis of photoprotective energy dissipation of excess energy. This study system allows valuable in-vivo characterization of photochemical processes without the need to resort to using isolated system with potential compromised responses.

This manuscript specifically focuses on the xanthophyll-cycle intermediate antheraxanthin that has been elusive and difficult to study. The manuscript reveals what promises to be an important function of antheraxanthin as a memory for prior exposure to excess light and an organism's ability to fully protect itself in rapidly fluctuating light environments that are so common in nature.

-The work supports the conclusions and claims made.

-The methodology is sound, although some of the specific details are not the primary area of my expertise. Enough detail is presented to allow the work to be reproduced.

We thank the reviewer for their comments and assessment of our communication and we hope the revisions we have made sufficiently address their specific comments below.

Specific comments

-Note that non-photochemical quenching per se dissipates only a very small fraction of excess excitation energy. The major pathway of non-photochemical energy utilization leads to thermal

energy and not to fluorescence emission. Avoid implying that non-photochemical fluorescence quenching is a major pathway of energy dissipation.

In the introduction to the model in Sec. II.A we have clarified explicitly that we assume the main dissipation channels are non-radiative. We have replaced " $k_{F,0}$ " with " $1/\tau_{F,0}$ " in Eq. (1) to clarify this and we have added the sentence "We assume that qE and qZ mechanisms are non-radiative, dissipating chlorophyll excitation energy as heat into the environment." after Eq. (1) to clarify this interpretation of our model.

-Consider defining HL as high light upon first mention and then consistently using the abbreviation for the rest of the text.

The initialisation HL is first defined in the second paragraph of the Sec. I, but it is now also redefined in Sec. II.A (second paragraph) where "HL" is used more extensively.

-The authors might consider elaborating on what had previously been reported on the topic of some sort of plant memory for excess light (examples of reviews touching on this subject include Plant Physiol. Rep. 2022, 27:560–572; Molecules 2020, 25, 5825).

We thank the reviewer for pointing us towards this important work on long-term memory of excess light exposure in plants. The following discussion of these studies and the appropriate references have been added to the third paragraph of the introduction (Sec. I):

"Studies on various plant species, including *Smilax australis*, *Monstera deliciosa*, *Vinca minor* and *Vinca major* have been shown to possess a long-term memory of growth light conditions, which is strongly species dependent. This long-term memory manifests in xanthophyll pool size and composition as well as maximum NPQ levels, an effect we also found evidence for previously in *N. Oceanica*."

-Concerning the elusive role of antheraxanthin in photoprotective energy dissipation, the authors could mention that correlations between chlorophyll fluorescence parameters and pigment levels reported for plants were higher when including A in addition to Z (see, e.g., Plants 1996, 198, 460-470).

Again we thank the reviewer for pointing us towards this work and it has now been referenced in the third paragraph of the introduction:

"The role of the partially de-epoxidised xanthophyll A in photoprotection has been difficult to investigate directly, however work on plants has suggested that both A and Z correlate with NPQ in plants."

-Where they comment on the relative contributions of qE and qZ in *N. oceanica*, the authors might consider mentioning that these features are impacted by species and growth light environment (different plant species, as well as plants of the same species grown in different light environments, vary widely in their maximal NPQ and in the proportions of qE and qZ; see,

e.g., Figure 2 in Plant Physiol. Rep. 2022, 27:560–572; Photosynthesis Research 2012, 113, 75-88; Planta 1998, 205, 367-374).

We have added a comment to this effect in the discussion (Sec. III) on this (paragraph 2), and the suggested references:

“It should be noted that this limit is based on extrapolating the model to light exposure times beyond those which we have investigated, which may not be reliable, and we also expect the relative contributions of qE and qZ to depend strongly on species and growth conditions, as has been found in studies of plants.”

Reviewer #2 (Remarks to the Author):

The manuscript “Kinetics of the Xanthophyll Cycle and its Role in the Photoprotective Memory and Response” presents a model and experimental data for how short term photoprotective memory works in the microalgae *Nannochloropsis oceanica*. I believe this is both novel and an exiting interpretation of how each part of the xanthophyll cycle impacts NPQ.

The work is original and will help researchers further dissect the contribution and kinetics of individual mechanisms that collectively result in the NPQ response. The way the xanthophyll's contribution to qE was resolved in this work is particularly interesting.

We thank the reviewer for taking the time to read and comment on our work. We hope the revisions we have made address the comments made below.

Concern that should be addressed before publication:

1) The authors state “We see rapid NPQ activation and deactivation in response to changes in light levels, occurring on a time-scale of less than 1 min, together with a slower increase in NPQ_{τ} during light exposure.” and this appears to be correct however, a quantification of activation/deactivation time would be good to be included here (e.g. decay half time).

This will also help support their next argument “We see the NPQ_{τ} recovers to its value at the end of the initial light period ($t = 5$ min) within 1 min in the second light period for dark durations up to $T = 5$ min, and even with a 20 min dark duration the NPQ_{τ} recovers within 3 min.”. Please provide some quantification of the NPQ activation time, or show those recovery points on the figures. A single exponential decay fitting works well on PAM derived NPQ, I expect that it would work with NPQt, see the following paper for such an example:

<https://doi.org/10.1038/s41598-019-44972-z>

From the model we have directly extracted the activation and deactivation rate constants - these are given in the supplementary information. We have now also added the following sentence, stating these rate constants in the discussion section as suggested:

“The rapid component of NPQ_{τ} activation and deactivation arising from the pH-sensing protein corresponds to the equilibration rate for the $PX \Leftrightarrow QX$ equilibrium in the model, given by $k_{QX}^{light/dark} = k_{QX,f}^{light/dark} + k_{QX,b}^{light/dark}$. This equilibration rate is 2.1 min⁻¹ under light conditions and 4.7 min⁻¹ in the dark which gives an activation time of 29 s and a deactivation time of 13 s.”

2) In Figure 3 A to G the only interesting part to visualise is the second illumination phase, which would be most interesting to see when all curves are combined in to one single plot. That way we can better assess the effect on NPQ induction of the different dark intervals. I suggest either modifying Figure 3 or to provide a supplementary figure where the above comparison is made.

We thank the reviewer for the suggestion. Combining the second illumination phase data as suggested is very cluttered and difficult to interpret. As an alternative we have added a supplementary figure comparing the averaged NPQ values (normalised by their value at $t = 5$ min) from the experiment and model in the first minute and the fourth minute of the second light period, as a function of the dark duration T . This figure is referred to in the main text in Sec. III, paragraph 5:

“In the supporting information, Fig.~S2, we show the experimental and model NPQ_{τ} recovery, averaged over the first minute of HL, in the second light phase for the 5 HL-T D- 5HL sequences, as a function of dark duration T . From this we have extracted (see SI for details) an NPQ_{τ} memory time-scale of ~ 22 min, which matches the model A \rightarrow V time-scale given by $1/k_{A \rightarrow V} = 19.9$ min. This strongly suggests that antheraxanthin acts as a short-term memory for light exposure, with the A \rightarrow V step of the xanthophyll cycle controlling the effective memory time-scale.”

3) In Figure 4 NPQ_t reaches negative values in multiple instances – please explain in the text why that is.

This arises due to noise in the fluorescence lifetime measurements. If $\tau_{F,meas}(t) > \tau_{F,meas}(0)$, then the experimental $NPQ_{\tau} < 0$. In the mutants NPQ is much closer to 0 than in the wild type, because one of the main quenching pathways has been deactivated in both, this means the noise is larger relative to the NPQ signal. This is unavoidable and does not concern us. The model predictions for NPQ_{τ} are however never negative, as expected.

4) I think some statistical evaluation of how well the model predicts the experimental data is needed (e.g. residual analysis, Chi-Square goodness-of-fit test, or something similar). I believe that would be needed for each individual figure where the model and experimental data are compared, including the figures with the training data.

All figure captions now included the root-mean square deviations (RMSD) values between the model predictions and data. Additionally in Fig. 3 we now include the R^2 values for the model predictions.

5) Please mention in the text the number of technical (if any) and biological replicates, as well as the number of experimental replication (how many times was this work repeated)?

For each sample sequence there were 3 technical replicates (indicated now in the captions for Fig. 3 and 4). The memory duration data was experimentally replicated twice as well as for the training data set. Only one experiment replicate was complete for the mutants, *vde* and *lhcx1*, with 2-3 technical replicates each. There were no biological replicates used for this paper.

6) Standard deviation is preferred to the standard error of mean

The standard error in the mean is the standard deviation in the mean of a number of measurements, and this is standard practice for reporting uncertainty in experimental measurements.

Minor observations:

1) This needs rephrasing “The accumulation of A and Z is correlated with increases in NPQ throughout a diurnal cycle in plants,10,11 and it has also been proposed that slower activating and de-activating quenching mediated by Z in the absence of a pH-gradient sensing protein, termed “qZ”, also operates in photosynthetic organisms.12,13”; qZ is not a pH sensing protein

Thank you for pointing out this unclear sentence. This has been rephrased to:

“The accumulation of A and Z has been observed to correlate with an increase in NPQ throughout a diurnal cycle in plants, and it has been proposed that a slower activating and deactivating Z-dependent quenching process also operates in the absence of a pH-gradient sensing protein, termed “qZ”.”

2) Make sure *N. oceanica* is in italic throughout the manuscript

This has been corrected in the revised version.

3) A few misspellings that need correction: “occurring”, “disprepancies”

Thank you for pointing these out - they have been corrected.

4) This is just an aesthetic suggestion: the white and yellow colouring that represent dark and light it's not ideal. The black and white bar system used in your previous paper is easier to follow (<https://doi.org/10.1063/5.0089335>)

This has been changed in the revised version of the manuscript.

Overall, I believe the manuscript presents novel and interesting findings, with significant implications for understanding the photoprotective memory in *Nannochloropsis oceanica*. Once the authors address the concerns outlined above, particularly with respect to providing statistical validation of the model predictions and more detailed quantification of the NPQ response time, I anticipate that this work will be a valuable contribution to the field. I look forward to seeing the revised manuscript.

Reviewer #3 (Remarks to the Author):

The manuscript deals with deciphering and modeling the kinetics of photoprotective xanthophyll pigments cycling (XC) and non-photochemical quenching (NPQ) in the microalga *Nannochloropsis*. The manuscript is well written, and the experimental data and modeling are convincing and of general value, i.e. XC and NPQ are photophysiological processes crucial to the photosynthetic productivity of most of all photosynthetic organisms, uni- and multi-cellular, aquatic and terrestrial. Nevertheless, there are, to my feeling, some lacks and unclear/uncomplete statements that require additional work and revisions before the manuscript could be published in *Nature Communications*. For these reasons, I suggest 'major revisions'. More detailed comments below.

Major comments:

-The nucleus of the model is based on the fact that a protein P binds a xanthophyll pigment (PX complex) and converts into a quencher complex (QX). This is fine. What is not is to consider that P is LHCX1 when we still do not know if LHCX1 (and any other LHCX protein) can effectively bind xanthophyll pigments, even more de-epoxidised forms, this questioning is even stated by the authors themselves in their introduction 'and whether or not LHCX1 can bind pigments is still under investigation'. Therefore, there is a central point I do not understand here, and which needs clarification. Furthermore, and based on this fact, I do not understand how (Part A-Results) 'the new model can quantitatively describe xanthophyll concentrations in cells, enabling us to estimate the absolute abundance of LHCX1': it would mean that it was demonstrated that LHCX1 effectively binds Zeaxanthin (Zx), which is not the case (or did I miss a recent report ?). There are some strong indications but no more so far (see ref 16 and 19 here).

We agree that LHCX1 has not been shown to bind pigments, and we have not claimed that LHCX1 alone acts as the quencher. LHCX1 clearly has a role in the PX/QX quenching process, as is shown by the *lhcx1* knock-out mutant NPQ data and modelling, however the model quencher QX may involve LHCX1 complexed with other light-harvesting proteins which do bind chlorophylls/xanthophylls, or LHCX1 may just induce conformational changes activating PX/QX. We do make several assumptions in obtaining our estimate of P abundance which are clearly outlined in the supporting information. One additional assumption is that each complex P has a single xanthophyll binding site, which we now clarify in Sec. II. The success of our model in interpreting the experimental data strongly indicates that LHCX1 is an essential component of qE. As such we feel our statement that "the new model can quantitatively describe xanthophyll concentrations in cells" is justified.

We have clarified the distinction between the model PX/QX and LHCX1 in Sec. II.A, adding "Previous work has identified LHCX1 as an essential component in activating the protein P, in the "qE" quenching mechanism, although the actual active quencher PX/QX could involve other proteins, especially since it is not known if LHCX1 binds pigments, and alternatively LHCX1 may

just induce the conformational changes in P to activate quenching. Thus the precise identity of PX/QX is open to interpretation.”

We have also clarified in Sec. II.A that we are assuming there is a single xanthophyll binding site per P, adding:

“For simplicity we assume a single labile xanthophyll binding site per P, and we have found that this is sufficient to interpret the available experimental data.”

We have also clarified that we only estimate the absolute abundance of quenching sites and not LHCX1.

-All along the manuscript, starting with the title, the authors claim about a ‘photoprotective memory’; how the authors came to this statement is unclear and the vocabulary used is itself obviously unclear to them: title and elsewhere in the manuscript (photoprotective memory), abstract (kind of memory, medium-term ‘memory’), part C of the Results (‘photoprotective memory’), etc. If the authors want to make this point the nucleus of their story (which I am not convinced it is), they should choose a wording and better explain this process. Also, if the authors want to strengthen this aspect, they should generalize to other organisms (see for instance <https://doi.org/10.3354/meps12398>) and to the in situ situation where the XC kinetics have been used to trace the water masses vertical mixing (numerous works by Brunet et al and by Bidigare et al, see for instance doi:10.1093/plankt/fbu069 ; doi:10.1093/plankt/fbr102 and references therein). This ‘memory’ aspect of the XC is not new and the authors should make better use of what has been published (even if not always the same time range) to strengthen their proposal.

We thank the reviewer for this observation. Most studies have focussed on the role of the xanthophyll cycle in much longer time-scale memory of light exposure, such as light exposure during growth and changes over the course of a day or longer. Here we are primarily concerned with short-term memory (up to ~1 hour), and directly connecting short-term xanthophyll kinetics with this short-term memory. We have added discussion in Sec. I, III and IV to this effect, clarifying that we are focussing on short-term memory - complementing existing research on long-term memory correlated with xanthophyll composition.

-Experiments: as much as the modeling versus experimental data is generally convincing (see minor comments below), I feel there are some obvious experiments that could be easily added to challenge the modeling procedure and, I am sure, strengthen and even more generalize the output of this work. I especially see two kinds of experiments (based on the HL-D sequences performed by the authors) : 1) the same kind of HL-D sequences but with sustained Zx (i.e. $qZ \neq 0$ at the beginning of the experiment), which could be, I guess, obtained by growing *Nannochloropsis* under a higher light intensity; 2) the same kind of HL-D sequences but replacing D phases by Low Light in order to speed up Zx epoxidation and NPQ relaxation (works by Goss et al, ref 11 here). $qZ \neq 0$ is especially paramount as more and more works report that in in situ conditions qZ is rarely 0 and there is often, if not always, sustained Zx and other de-epoxidized xanthophylls.

Whilst we agree that these experiments would be very interesting to perform and plan to perform them in future work, we feel that adding these experiments and the associated modeling to the current paper is beyond the scope of this communication.

Minor comments:

Abstract:

-‘quantitative model we show’: dot missing between model and we show.

Thank you - this has been corrected.

-last sentence: why only vascular plants ? There is a growing blue biotechnology field, mostly based on marine microalgae (among which *Nannochloropsis*) that would be interested in such a work.

We have added “and algal” to the final sentence of the abstract.

Introduction:

-*N. oceanica* in italics.

Thank you - this has been corrected.

-‘a pH-sensing protein LHCX1’: this is contrary to current knowledge and to the first sentence/2nd column of the abstract-page 1 (statement ‘still under investigation’), indeed we do not know yet what exactly is the role of LHCX1, and there are recent works even suggesting it does not sense pH at all (see ref 16 and 19 here).

We have rephrased this sentence to reflect the current uncertainty in the role of LHCX1.

“It consists of two main components: a pH-sensing protein, potentially LHCX1, and the xanthophyll cycle.”

-‘more complex features’: I do not understand why state-transitions are a ‘more complex feature’ than the XC and NPQ, this is wrong.

Here we mean features which are more complex to incorporate into our biochemical kinetic model, as such we have rephrased this to “additional features”.

Results:

-HPLC data-Figure 2: units, is ‘Chl’ Chlorophyll a ? Because these data are normalized over Chl a (I guess), it is needed to show that during all experiments the Chl a content/cell does not change; regarding the length of the experiments, I guess there is no change but still this is important to confirm.

Thank you for pointing this out. Yes the units are mmol (Car)/ mol Chl a since *Nannochloropsis* does not contain any other forms of chlorophyll. This is corrected in the text in the Fig 2 caption. For our HPLC procedure, we try to utilise approximately the same amount of cells for each sample; however, during the processing of the algal cells, cells can be lost in transfers. Therefore we normalise to Chl a concentration rather than cell count to ensure the pigment concentrations can be accurately compared. Additionally, the Chl a concentration is not

expected to change significantly over the course of 20 minutes, allowing us to normalise to Chl a content (Lichtenthaler, H. K. & Babani, F. Chlorophyll a Fluorescence, A Signature of Photosynthesis. *Adv. Photosynth. Respir.* 713–736 (2004) doi:10.1007/978-1-4020-3218-9_28.).

-page 4, column 1, top: ‘the model accurately predicts’, no, this is an overstatement that needs to be modulated; the model works fine to predict a trend, especially when the light climate is rather stable (Figure 2) but it does not work so well in ‘accurately’ predicting changes in XC dynamics when light fluctuates (Figure 2 D).

Careful inspection reveals that the fluctuations do not correlate with the 2 min periodicity, thus we conclude that these fluctuations are just noise. With the exception of 2 B, where the model over predicts A and Z production, the model is within the experimental fluctuations. We have rephrased this to say:

“we see the model mostly predicts the HPLC data within the experimental error, although in the 1 HL- 4 D- 7 HL- 5 D- 1 HL- 2 D sequence the model slightly overestimates $\Delta[A]$ and $\Delta[Z]$ after 1 min of light exposure (it should be noted that the fluctuations in xanthophyll concentrations in Fig. 2 D do not correlate with the periodicity of light exposure on close inspection).”

-Part D-Results: there is no information on the mutants: we can guess these are KO mutants, but otherwise we do not know how they were produced, especially from this work or from another piece of work, and if from a previous work, there is no reference.

This has now been clarified in the methods section. The following sentence has been added:

“The knock-out mutants *vde* and *lhcx1* were generated using the same procedure as in Ref. [Park2019], wherein homologous recombination of a hygromycin resistance cassette was used, with the addition of Cas9 mediated RNP for *lhcx1*.”

-Part D-Results, last paragraph: KO LHCX1 mutant, there is no explanation (in the corresponding Discussion) on how NPQ induction is feasible in such mutant; see ref 15 here.

We have not suggested a specific microscopic model for qZ. As explained in Sec. II.A, we model it by adding a quenching process proportional to the pool [Z] concentration. In the model for the *lhcx1* mutant we only remove “P” from the model, so qZ is still included. In the supporting information we have added a section describing how a second protein or protein complex P’ of unknown identity binding Z could be consistent with our simple model for qZ. We cannot say any more about qZ from this study. In the discussion we have added the following sentence referring to this:

“We have not suggested a microscopic model for the qZ process, although in the SI, Sec. S.4, we show how a quenching process depending on some other zeaxanthin binding protein (or complex) P’ would be consistent with our simple model. Zeaxanthin binding to some other protein could activate qZ by directly quenching excitation energy, potentially via charge transfer, or inducing conformational changes in the protein that promote other quenching mechanisms.”

Discussion:

-bottom page 5: 'poor quencher, modest quencher and good quencher', and just below A is not anymore depicted as a 'modest' quencher but as an 'intermediate' one; these qualitative subjective statements should be avoided.

From our model we can directly quantify the NPQ capacity of each xanthophyll (see Sec. II.C) so the statements poor, modest and intermediate are quantified. The use of the term intermediate here refers to the fact that A is the intermediate species between V and Z in the VAZ cycle. We realise how this can cause confusion, so we have changed this to "modest quencher".

Conclusion:

-'qZ contribution to NPQ being much smaller than the one of qE': no, it depends on species and on environmental conditions; see the works by Demmig-Adams et al., Verhoeven et al and Lacour et al. on sustained qZ-like NPQ in tropical and desert plants, evergreen conifers and polar diatoms.

We have added a comment to this effect in the discussion (Sec. III) on this (paragraph 2), and the suggested references:

"It should

be noted that this limit is based on extrapolating the model to light exposure times beyond those which we have investigated, which may not be reliable, and we also expect the relative contributions of qE and qZ to depend strongly on species and growth conditions, as has been found in studies of plants."

MandM:

-Part C-HPLC: 'Standard curves from concentrated pigments', I do not understand what is meant by 'concentrated pigments' ?

Using samples with known concentrations of certain pigments, we can use this to help identify and quantify through both retention times and spectral characteristics which pigments are described by the HPLC peaks.

List of references:

-species names in italics.

-some references need to be updated (for ex. 14).

Thank you for pointing out these issues - they have been corrected.

Reviewer #2 (Remarks to the Author):

I am satisfied with how my concerns have been addressed.

Reviewer #3 (Remarks to the Author):

The authors well answered my requests, I congratulate them for this very nice piece of work, and I now support publication.

Johann Lavaud